# The Gene Expression Landscape of Prostate Cancer BM Reveals Close Interaction with the Bone Microenvironment

**DOI:** 10.3390/ijms232113029

**Published:** 2022-10-27

**Authors:** Alireza Saraji, Kang Duan, Christian Watermann, Katharina Hempel, Marie C. Roesch, Rosemarie Krupar, Janine Stegmann-Frehse, Danny Jonigk, Mark Philipp Kuehnel, Wolfram Klapper, Axel S. Merseburger, Jutta Kirfel, Sven Perner, Anne Offermann, Verena Sailer

**Affiliations:** 1Institute of Pathology, University Hospital Schleswig-Holstein, Campus Luebeck, 23538 Luebeck, Germany; 2Department of Urology, University Hospital Schleswig-Holstein, Campus Luebeck, 23538 Luebeck, Germany; 3Research Center Borstel, Leibniz Lung Center, 23845 Borstel, Germany; 4Institute of Pathology, Hannover Medical School, 30625 Hannover, Germany; 5Institute of Pathology, Hematopathology Section and Lymph Node Registry, Universitätsklinikum Schleswig-Holstein, Campus Kiel, 24105 Kiel, Germany

**Keywords:** BM, prostate cancer, transcriptome, tumor microenvironment

## Abstract

Bone metastatic (BM) prostate cancer (PCa) belongs to the most lethal form of PCa, and therapeutic options are limited. Molecular profiling of metastases contributes to the understanding of mechanisms defining the bone metastatic niche. Our aim was to explore the transcriptional profile of PCa BM and to identify genes that drive progression. Paraffin-embedded tissues of 28 primary PCa and 30 BM were submitted to RNA extraction and analyzed by RNA sequencing using the Nanostring nCounter gene expression platform. A total of 770 cancer-related genes were measured using the Nanostring™ PanCancer progression panel. Gene Ontology (GO), KEGG, Reactome, STRING, Metascape, PANTHER, and Pubmed were used for data integration and gene annotation. We identified 116 differentially expressed genes (DEG) in BM compared to primaries. The most significant DEGs include *CD36*, *FOXC2*, *CHAD*, *SPP1*, *MMPs*, *IBSP*, and *PTX3*, which are more highly expressed in BM, and *ACTG2*, *MYH11*, *CNN1*, *FGF2*, *SPOCK3*, and *CHRDL1*, which have a lower expression. DEGs functionally relate to extracellular matrix (ECM) proteoglycans, ECM-receptors, cell-substrate adhesion, cell motility as well as receptor tyrosine kinase signaling and response to growth factors. Data integration and gene annotation of 116 DEGs were used to build a gene platform which we termed “Manually Annotated and Curated Nanostring-data Platform”. In summary, our results highlight the significance of certain genes in PCa BM to which essential pro-metastatic functions could be ascribed. Data from this study provide a comprehensive platform of genes that are related to PCa BM and provide evidence for further investigations.

## 1. Introduction

Prostate cancer (PCa) is the most common non-cutaneous malignancy in men with estimated 248,530 new cases in 2021 in Western countries [1]. The majority of patients will be cured of their disease but a significant number of men will experience disease progression and ultimately succumb to metastatic PCa. A total of 34,130 deaths from PCa in the United States are projected for 2021. Approximately 5% of patients with PCa present with de novo metastatic disease [2]. Metastatic PCa displays a remarkable osteotropism and—usually osteoblastic—skeletal metastases are found in more than 90% of late-stage PCa [3]. BM results in high morbidity and skeletal-related events such as fractures or pain and can significantly impair a patient’s quality of life. It has now been recognized that the bone microenvironment plays a significant role in facilitating the seeding of PCa cells to the bone and that the bone metastatic (BM) niche is a unique habitat [4]. The interaction between bone stromal cells and PCa cells is critical in establishing metastatic lesions. Characterizing both PCa BM and the bone microenvironment can contribute to understanding the intricate biology of PCa BM formation and might help improve the clinical management of these patients. Molecular studies can help to shed light on this interplay and have already been widely employed in the setting of metastatic PCa [5,6,7]. This effort is particularly important because we believe there is an unmet need for those patients whose cancer has progressed and spread beyond the prostate. While most molecular insight from PCa BM has been gained in large sequencing studies, it is required to implement our knowledge into everyday practice to ultimately improve patient care. Tissue acquisition is one of the foremost obstacles that have to be overcome in studying PCa BM. Biobanking protocols have been established to perform next-generation sequencing from BM using fresh frozen tissue [8]. However, the majority of available BM tissue will be obtained via routine surgery and will thus often be decalcified formalin-fixed and paraffin-embedded (DFFPE). We aimed to study the transcriptome of PCa BM, thus shedding light both on the tumor cells as well as on the tissue-specific microenvironment. Methodologically, in our previous study [9] as well as the current one we show that even several years old archival materials can be used for digital gene expression analysis using the NanoString^®^ platform on PCa BM and primary PCa. The delicate balance of gene expression in a highly specialized environment such as the bone might give us an insight into the formation of metastatic lesions, in particular in comparison with patterns of gene expression in the primary tumor [10].

## 2. Results

### 2.1. Differentially Expressed Genes (DEGs) in PCa BM Compared to Primary Tumors

A schematic overview of the experimental study design is provided in (Figure 1). By comparing the RNA expression level of 770 genes in our samples, we found a total of 116 genes with significantly different expression in PCa BM compared to localized primary tumors. All DEGs are listed and annotated according to the literature and bioinformatic platforms, as described above, regarding PCa and BM (**MACNP**, Appendix A). Gene clustering according to the tissue type reveals both gene subsets to be specifically up-regulated in BM or primaries as well as subgroups within the BM and primary tumors characterized by marked up- or down-regulated gene sets (Figure 2A). Genes with the highest fold change and most significant *p*-value between BM and primary tumors include *CD36*, *FOXC2*, *COL1A2*, *CHAD*, *SPP1*, *COL1A1*, *MMP9*, *IBSP*, *PTX3*, and *MMP13* which are more highly expressed in BM, and *ACTG2*, *MYH11*, *MFAP4*, *ITGA8*, *CNN1*, *FGF2*, *ISL1*, *SFRP1*, *SPOCK3* and *CHRDL1*, which have a lower expression in BM (Figure 2B). The top 10 genes with the highest fold change are listed in (Figure 2C) (genes that are lower expressed in BM) and (Figure 2D) (genes that are higher expressed in BM).

Additionally, to evaluate whether circulating tumor cells (CTCs) provide similar results as tumor tissue, we collected liquid biopsy blood samples from 15 patients with PCa BM. A total of 5 out of the 15 patients were identified as CTC-positive. From these patients, mRNA was purified and isolated from the CTC-positive samples, however, due to the minute amount of isolated mRNA (average mRNA concentration < 11 ng/µL; average percentage of mRNA content above threshold < 50%) nSolver^®^ was not able to analyze the data compared to the primary tumor. To avoid this limitation in the future, an amplification step by using the nCounter^®^ Low RNA Input Kit (NanoString, Seattle, WA, USA) is suggested.

### 2.2. Functional Assignment of Top DEGs

The top 20 DEGs were assigned to molecular functions using Metascape. The strongest association of these genes was found to include extracellular matrix (ECM) proteoglycans, ECM-receptor interaction, and elastic fiber formation (Figure 3A). DEG with the highest fold change and significance are functionally associated with extracellular structure organization and blood vessel development, which also include most genes (gene count, reflected by bubble size) (Figure 3A). Other BM-related features which were linked to DEG include regulation of smooth muscle cell proliferation, ossification, cell-substrate adhesion, positive regulation cell motility, receptor tyrosine kinase signaling, response to wounding, fiber organization, and response to growth factors (Figure 3A). Gene enrichment analyses designating certain functional features included the top 20 DEG showing both up-regulation and down-regulation in BM compared to primary tumors (Figure 3B).

### 2.3. PPI and Common Functional Assignments of Top DEGs

DEG was further used for PPI analysis using String. We found a high degree of interactions between up- and down-regulated genes with *FN1*, *MMP9*, *COL1A1*, *COL1A2*, and *FGF2* showing the highest degree of global network interactions. In addition, several DEG showed overlapping molecular functions, revealing that cell motility includes the highest number of overlapping DEG followed by cell adhesion and ECM proteoglycans (Figure 4). Eight DEG have overlapping functions referring to the regulation of cell motility and cell differentiation, while seven genes have overlapping functions referring to the regulation of cell motility, cell differentiation, and cell adhesion (Figure 5A). *COL3A1*, *COL1A1*, *FN1*, *CX3CL1*, *SFRP1*, *THY1*, *NRP1*, *SEMA3*E, *POSTN*, *GREM1*, and *ITGB3* are associated with multiple metastasis-related features (Figure 5B). *ITGB3* can be functionally linked to ECM proteoglycans, cell differentiation, adhesion, and cell motility which are key features during metastasis (Figure 5B).

## 3. Discussion

Metastasis to the bone requires certain features of cancer cells which are able to spread from their primary localization to distant sites, to survive in and adapt to a new microenvironment and to proliferate which enables metastatic tumor growth [11]. The molecular characterization of these cancer cells contributes to a deeper understanding of biological processes during metastasis that may lead to the identification of predictive biomarkers for metastasis, and to the identification of novel therapeutic targets. In order to explore the transcriptome of BM PCa cells reflecting their phenotype on the gene transcriptional level, we compared the expression of 770 cancer-related genes in 30 BM PCa and 30 primary PCa.

Overall, we observed that most DEGs are associated with cellular features that are essential for metastatic spread. These include cancer cell-intrinsic features such as cell-substrate adhesion and cell motility as well as genes reflecting the interaction between cancer cells and the tumor microenvironment (Figure 3A,B). The latter comprises genes referring to ECM proteoglycans, ECM-receptor interaction, and extracellular structure organization, which are essential components for the initiation and propagation of BM [11].

In more detail, we found collagen *(COL) 1A1* and *COL1A2* as one of the most significant up-regulated genes in BM which is in line with previous studies reporting *COL1A1* and *COL1A2* to be associated with an aggressive and pro-metastatic phenotype in diverse cancer types [12,13,14]. Notably, high levels of *COL1A1* and *COL1A2* have been observed to contribute in particular to PCa BM [15] and castration resistance [16], which supports our findings. Furthermore, we observed increased levels of *MMP9* and *MMP13* in BM, both belonging to the family of proteinases with functions in ECM remodeling [17]. The role of *MMP9* in metastasis has been reported to be diverse, but its up-regulation is associated with poor survival, e.g., in breast cancer patients [18]. Functional and in vivo studies observed both pro-metastatic as well as metastasis-preventing functions of *MMP9*, potentially dependent on the cellular and genetic background [19,20]. In principle, *MMP9* is secreted by tumor-associated fibroblasts and cancer cells. In line with our findings showing a highly significant up-regulation of *MMP9* in BM, *MMP9* positively affects PCa cell invasiveness [21].

Interestingly, we found a highly significant up-regulation of *CD36* in BM, supporting previous findings on important pro-tumorigenic features of *CD36* [22]. *CD36* is a scavenger receptor for fatty acid uptake that affects lipid metabolism, adhesion to the ECM, TGFβ activation, immune signaling, and platelet activation [23]. *CD36* contributes to ovarian cancer metastasis [24], promotes epithelial-mesenchymal transition (EMT) in cervical cancer via TGFβ signaling [25], and accelerates gastric cancer metastasis through reprogramming lipid metabolism [26], highlighting its broad involvement in cancer progression. Recently, Watt et al. identified *CD36* to be an essential regulator of lipid metabolism in patient-derived PCa xenograft mouse models, thereby promoting *CD36*-dependent PCa cancer progression [27]. Since publications on *CD36* expression in PCa are lacking so far, our study foremostly identifies *CD36* up-regulation in PCa BM to stimulate further investigations.

Other highly expressed DEGs with known implications for PCa include *FOCX2*, which is highly expressed in BM PCa and co-occurs with an EMT phenotype [28], the cell adhesion molecule *SPP1* encoding osteopontin [29,30], and Pentraxin-3 (*PTX3*) that has emerged as a predictive marker for a high risk of PCa development [31,32]. 

Out of the most significantly down-regulated genes in BM, tumor suppressive implications have been reported for myosin heavy chain 11 (*MYH11*) in gastric cancer [33] and integrin subunit α 8 (*ITGA8*). Interestingly, a tumor-suppressive role of *ITGA8*, mainly through *ITGA8* silencing by gene hypermethylation, has been reported in breast cancer [34] and renal cell carcinoma [35]. High levels of *SPOCK3* have been reported to correlate with a better outcome in PCa patients [36]. *SPOCK3*, also called osteonectin, is a member of the calcium-binding proteoglycan protein family and might mediate its tumor suppressive function in PCa by its suppressive effect on tumor invasion [37].

In summary, our study provides data on the transcriptional profiles of primary PCa compared to PCa BM with subsequent comprehensive bioinformatic analysis and integration of the current literature. Therefore, our data provide evidence for further functional exploration that is essentially needed to develop certain genes as biomarkers and therapeutic targets. Notably, we present the Manually Annotated and Curated Nanostring-data Platform (MACNP) which includes detailed and bioinformatically integrated information about the most significant 116 DEG. This platform might be used by researchers to integrate their own data and to identify candidate genes for experimental exploration. Overall, our data confirm previous knowledge about gene groups and certain genes to be important for PCa BM, and highlight promising candidate genes such as *CD36*, *FOCX2*, and *PTX3* as pro-oncogenic molecules for the development of BM. 

## 4. Materials and Methods

### 4.1. FFPE Specimens and Cohort Description

This study included a cohort of 58 patients: DFFPE specimens from 28 patients with primary PCa and 30 patients with PCa BM, which were obtained by radical prostatectomy or transurethral resection of the prostate (TUR-P). Radical prostatectomy specimens were collected from the archive of the Institute of Pathology, Hospital of Goeppingen, Germany. BM was collected from the archive of the Institute of Pathology, University Hospital Schleswig-Holstein (UKSH), Campus Luebeck, Germany, and the Institute of Pathology, Hematopathology Section and Lymph Node Registry, University Hospital Schleswig-Holstein (UKSH), Campus Kiel, Germany. Histopathological evaluation and annotation for microdissection were performed by two pathologists (V.S. and A.O.)

### 4.2. RNA Extraction from Decalcified FFPE Specimens

FFPE specimens were sectioned into 8 μm thick cuts and two tissue sections were placed on each slide. Sections were compared with the annotated HE slide and only the marked tumor tissue was scraped off with a scalpel and transferred directly into the lysis buffer in an RNAase-free tube. RNA was isolated using the automatic bead-based Maxwell^®^ RSC RNA FFPE Kit (Cat. No: AS1440, Promega, Madison, WI, USA) according to the manufacturer`s manual and guidance. The RNA was eluted in water and then measured with NanoDrop^®^ or Qubit^TM^ (London, UK). The RNA samples were divided into 7 μL aliquots and stored at −80 °C.

### 4.3. mRNA Analysis (Digital Analysis for Transcriptomes)

mRNA expression profiling was performed using the Nanostring nCounter gene expression platform (Nanotring Technologies, Seattle, WA, USA). To evaluate cancer-related genes, we applied Nanostring™ PanCancer progression panel comprising 770 genes. The samples were loaded (10–35 ng RNA in a total of 30 μL loading mixture) on a special cartridge and we proceeded by utilizing a fully-automated Prep Station following the manufacturer’s recommendations (Nanostring™ Inc.). The proceeded cartridge was then sent for digital analysis with the nCounter^®^ Sprint Profiler system (performed at the Institute of Pathology, Hannover Medical School, Hannover, Germany). The data were exported as reporter code count (RCC) files and imported to the Nanostring nSolver™ analysis software v4.0 for further analysis. Automatic quality control of mRNA was performed according to the software instructions.

### 4.4. DEGs Analysis

The raw mRNA counts were analyzed by nSolver according to the guidelines of the nSolver analysis software. After removing samples with the quality control (QC) flag, mRNA count normalization and log_2_ Fold Change (FC) calculation with Welch’s *t*-test *p*-value were performed. Significant DEGs were defined by *p* < 0.05 and |log_2_FC| > 1 as the threshold. R language (R version 4.0.3) was used to generate the figures.

### 4.5. Gene Annotation (GA) and Biological Themes Enrichment Analysis

All DEGs were loaded into Metascape (metascape.org) to perform GA and enrichment analysis. DEGs’ biological themes enrichment includes GO terms, KEGG pathway, Reactome, etc. All enrichment terms took *p* < 0.05 as the cut-off threshold and the summary of the top 20 collections was displayed. 

### 4.6. Construction of Protein–Protein Interaction (PPI) Network and Hub Proteins Determination

To explore the internal interactions between DEGs, we used STRING to construct the PPI network. A confidence score of ≥0.4 was considered the threshold of significance. PPI network tabular text was further imported into Cytoscape (version 3.8.2) software, and the cytoHubba app was used for hub proteins computing. Hub proteins were calculated by the MCC (maximal clique centrality) method and marked with red to light yellow according to MCC values.

### 4.7. Data Platform Creation (MCNP Build Up)

The MACNP stands for “Manually Annotated and Curated Nanostring-data Platform” for 116 DEGs (PCa BM vs. primary PCa). Data output obtained from the nSolver™ analyzer software was manually annotated and enriched for up- and down-regulated DEGs based on the following criteria and tools:

(I) Metascape for molecular function enrichment (metascape.org), (II) PANTHER for biological processes enrichment (pantherdb.org), (III) protein family using Pfam database (pfam.xfam.org), (IV) pathway enrichment retrieved from KEGG (genome.jp/keg) and (V) extracting publication for each gene by using related keywords (pubmed.ncbi.nlm.nih.gov (accessed on 15 October 2020)). 

### 4.8. Liquid Biopsy (Circulating Tumor Cells (CTC) Detection and mRNA Isolation) 

Liquid biopsy was performed using AdnaTest (Quiagen, Hilden, Germany). Briefly, peripheral blood (7–8 mL whole blood) from patients with PCa BM was collected into EDTA collection tubes. According to the manufacturer’s protocol AdnaTest ProstateCancerSelect (Quiagen, Hilden, Germany) with the help of an antibody-conjugated magnetic bead, CTCs were purified and enriched followed by mRNA extraction step. Furthermore, following the manufacturer’s protocol for AdnaTest ProstateCancerDetect (Quiagen, Hilden, Germany), the isolated mRNA was subjected to a multiplex RT-PCR of different PCa-associated markers. The expression patterns of CTCs were then analyzed using a bioanalyzer (Agilent^®^ Technologies, Inc. Santa Clara, CA USA). Finally, the extracted mRNA from CTCs were loaded on the nCounter^®^ cartridge and then we proceeded with the nCounter^®^ Sprint Profiler system, as described before.

## Figures and Tables

**Figure 1 ijms-23-13029-f001:**
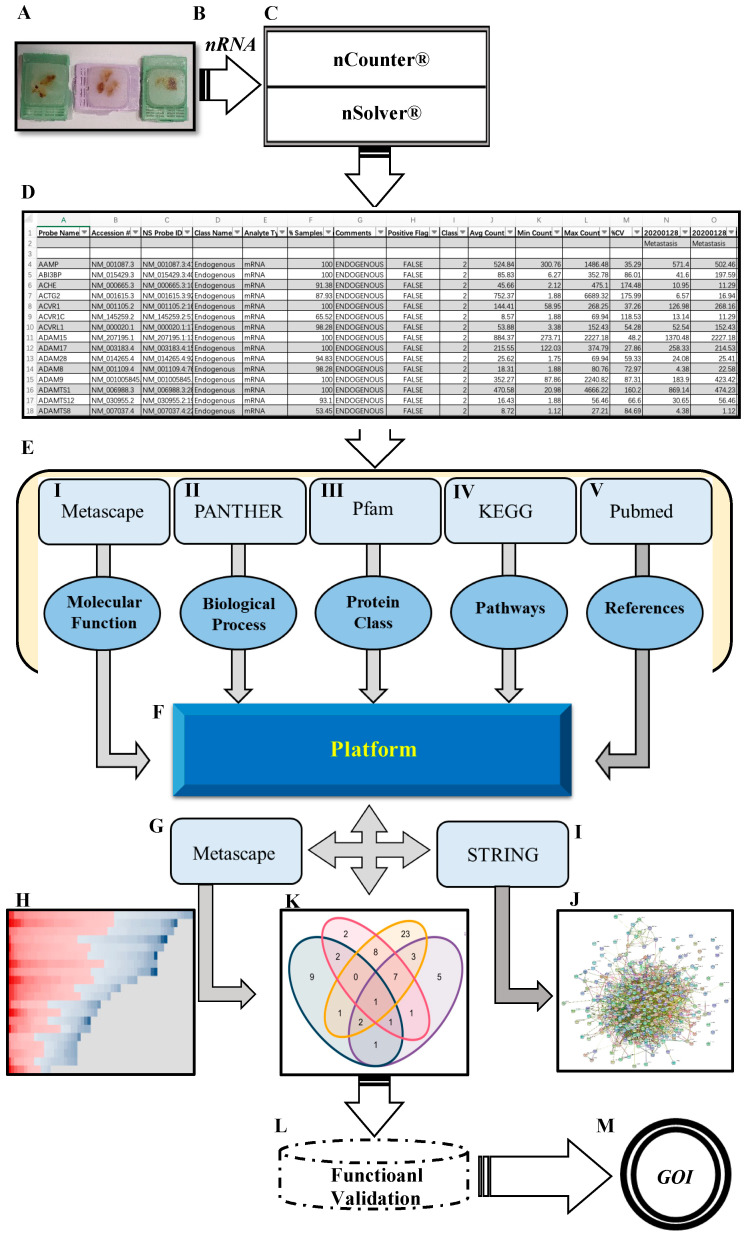
**Schematic overview of the workflow.** (**A**) FFPE tissue from PCa BM patients. (**B**) High quantity and quality of mRNA extracted from FFPE. (**C**) Fully-automated gene expression assay utilizing nCounter™ digital analyzer followed by nSolver™ NanoString technology v4.0. (**D**) Data output obtained from nSolver™ analyzer software. (**E**) Manual annotation and data enrichment procedure of DEGs: (I) Metascape for molecular function enrichment (metascape.org), (II) PANTHER for biological processes enrichment (pantherdb.org), (III) protein family using Pfam database (pfam.xfam.org), (IV) pathway enrichment retrieved from KEGG (genome.jp/keg), and (V) extracting publication for each gene by using related keywords (pubmed.ncbi.nlm.nih.gov). (**F**) The MACNP stands for Manually Annotated and Curated Nanostring-data Platform for 116 DEGs (PCa BM vs. primary PCa). (**G**,**H**) Further enrichment and clustering DEGs based on the interaction between MACNP platform and Metascape. (**I**,**J**) Protein–protein interaction (PPI) of DEGs analysis produced by String (string-db.org) and Cytoscape (Cytoscape.org). (**K**) Venn diagram representing overlapping genes obtained from the interaction between MACNP platform and bioinformatics.psb.ugent.be/webtools. (**L**,**M**) Further research schedule.

**Figure 2 ijms-23-13029-f002:**
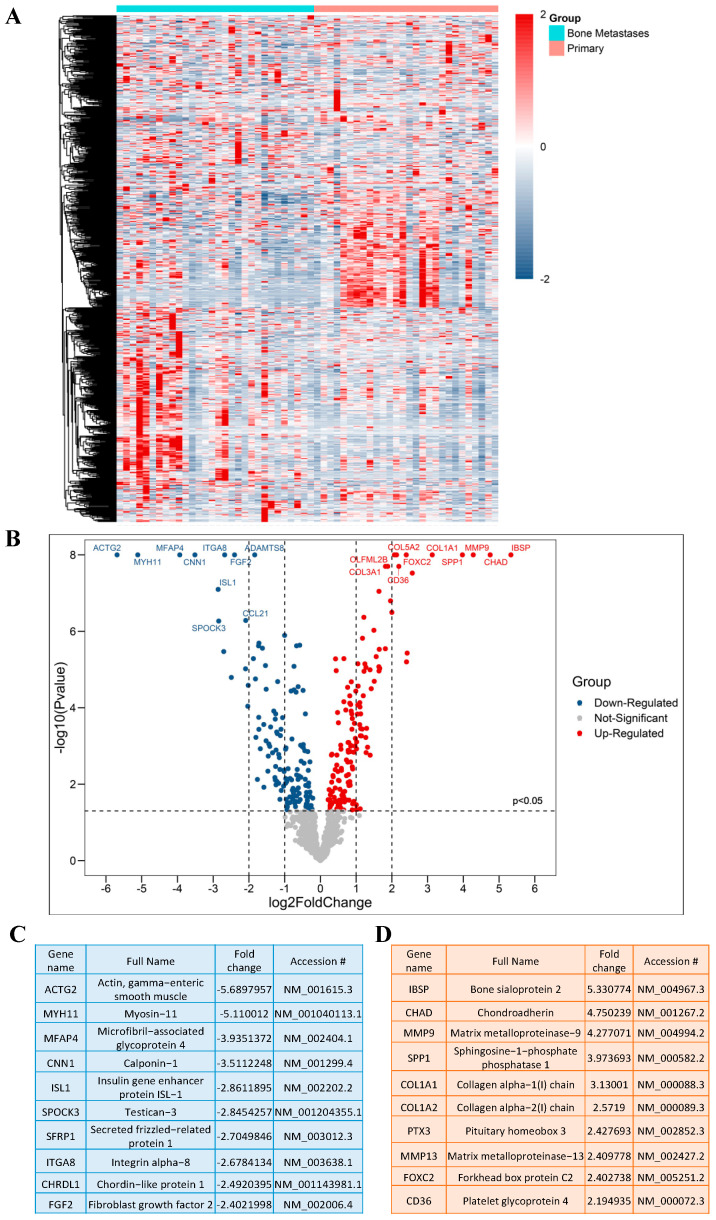
**DEGs in PCa BM.** (**A**) Heat map representing DEGs, produced by “Morpheus”. (**B**) Volcano plot of significant DEGs based on the *p*-value and fold change; Up indicates upregulated and Down indicates downregulated genes. (**C**) Top 10 significantly downregulated DEGs based on the fold change. (**D**) Top 10 significantly upregulated DEGs based on the fold change.

**Figure 3 ijms-23-13029-f003:**
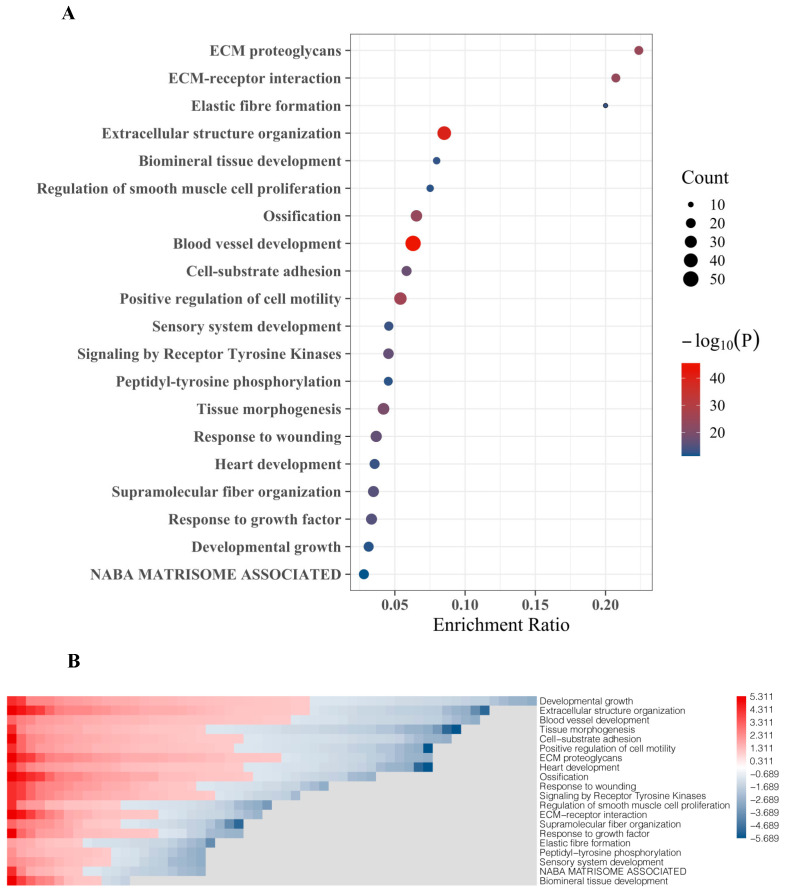
**Gene ontology (GO) and functional enrichment of top 20 gene set involved in PCa BM.** (**A**) Bubble chart representing the top 20 DEGs set enrichment based on their molecular activities obtained from Metascape. (**B**) Heat map representing GO and biological processes enrichment of DEGs obtained from Metascape and GO.

**Figure 4 ijms-23-13029-f004:**
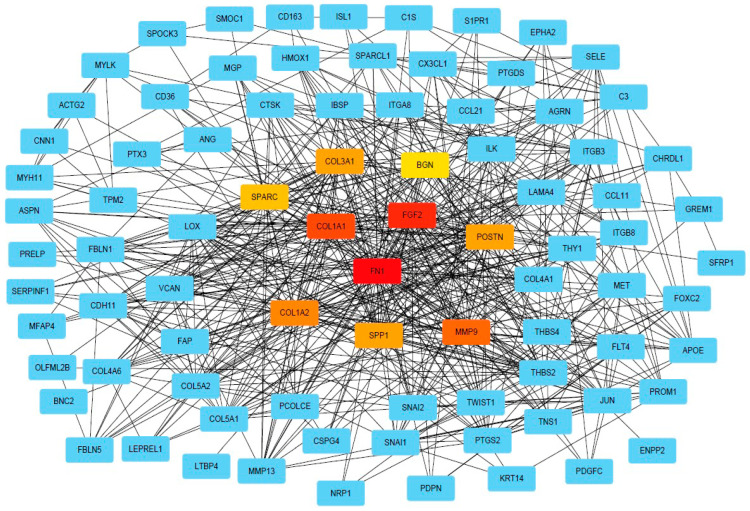
**PPI network of DEGs.** PPI network of DEGs analysis provided by String (string-db.org) and Cytoscape (Cytoscape.org), squares indicate DEGs, and lines indicate interactions. The intensity in red of the squares indicates the degree of global network interactions.

**Figure 5 ijms-23-13029-f005:**
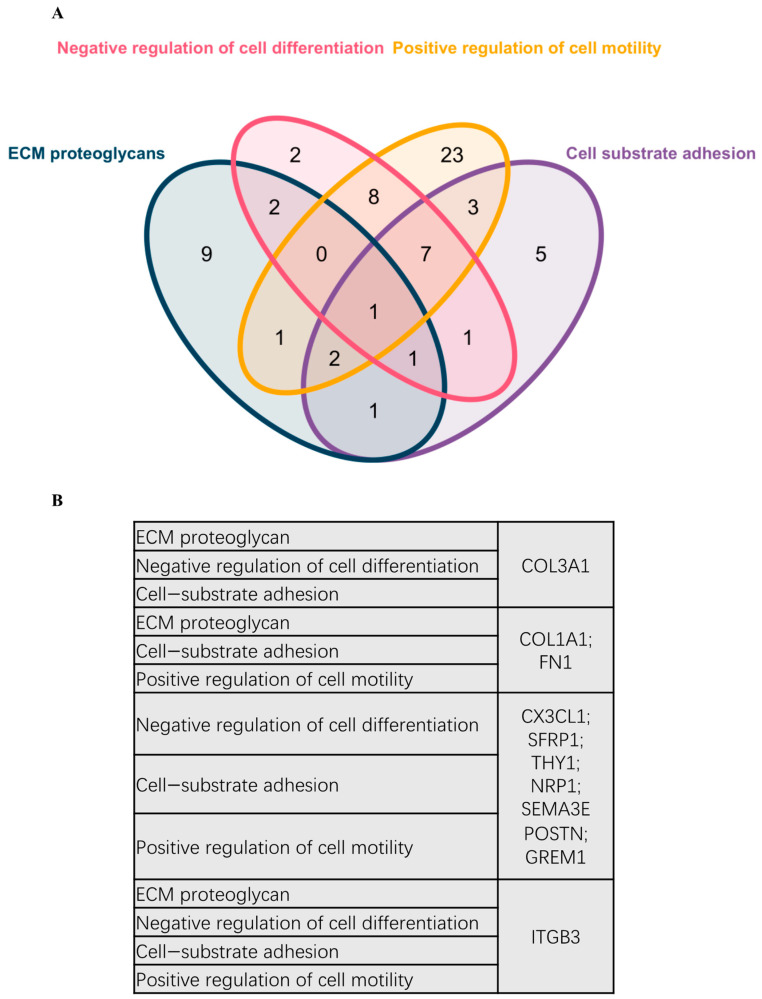
**Venn diagram representing overlapping DEGs in different pathways and biological processes.** (**A**) Venn diagram representing overlapping DEGs in different pathways and biological processes. (**B**) Scoring references list representing the detailed DEGs involved in the Venn diagram.

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
