# Peer review of "The Gene Expression Landscape of Prostate Cancer BM Reveals Close Interaction with the Bone Microenvironment"

_ijms, 2022, doi:10.3390/ijms232113029_

Round 1

Reviewer 1 Report

Offermann et al. conducted a transcriptome profiling study of 28 primary PCa tumors and 20 metastases using an RNA-seq approach. As a result of the study, a number of differentially expressed genes associated with PCa metastases were identified, as well as the main biological pathways. As part of the study, transcriptome profiling of circulating tumor cells was also carried out, however, due to technical difficulties, it was not possible for the authors to analyze the obtained data in comparison with primary tumor samples. managed. However, before a manuscript is considered suitable for publication, several issues need to be resolved:

Major notes:

 - The authors used a two-tail T-test to assess statistical significance. Are the data in the comparison groups under consideration normally distributed? Perhaps the authors should consider the use of non-parametric analogues.

- It would be interesting to discuss the expression of identified genes in the progression of prostate cancer in general, for example, based on bioinformatic analysis of public data, such as TCGA.

Minor remarks:

- Authors should adhere to the introduced abbreviations within the text (prostate cancer - PCa).

- There are many typos in the text.

- In the materials and methods, it is worth specifying in more detail the tools used, for example, for working in the statistical environment R.

Author Response

Dear reviewer 1:

Reviewer 2 Report

In this study, the author aimed to explore the transcriptional profile of prostate cancer bone metastatic and identify genes that drive progression. According to differentially expressed genes, they explored their function and how they regulate the prostate cancer bone metastatic process. However, there are some major points to improve.

1.      The author only used their own RNA-seq data, I suggest the author could also use the TCGA or GEO database to do further verification. The TCGA database is a pan-cancer RNA-seq database that included multiple types of cancer patients’ information (RNA-Seq and clinic data). You could use the Cbioportal website to get access to these data.

2.      In the functional enrichment analysis, the author only used KEGG and GO analysis. I suggest the author could also try hallmark gene sets, which are cancer specific.

3.      The prognostic exploration is a really important part of cancer research; I’d be really happy to see if the author could have a look at the prognostic and clinic value of these DEGs.

Author Response

Dear reviewer 2:

Round 2

Reviewer 2 Report

The authors have answered all of my questions and the paper has been greatly improved.